# Bridges or Barriers? Cross-boundary communication and governance mismatches in co-managed protected areas

Rebecca Borges[1,2,3]*, Roberta Sá Leitão Barboza[3,4], Theresa Schwenke[5,6]

1 Helmholtz Institute for Functional Marine Biodiversity at the University of Oldenburg (HIFMB), Oldenburg, Germany, 2 Alfred-Wegener-Institut Helmholtz-Zentrum für Polar- und Meeresforschung, Bremerhaven, Germany, 3 Laboratório de Ensino, Pesquisa e Extensão Pesqueira junto a Comunidades Amazônicas (LABPEXCA), Instituto de Estudos Costeiros (IECOS), Federal University of Pará (UFPA), Bragança, Pará, Brazil, 4 Diretoria de Pesquisas Sociais, Fundação Joaquim Nabuco, Recife, Pernambuco, Brazil, 5 Leibniz-Zentrum für Marine Tropenforschung (ZMT), Bremen, Germany, 6 Universität Bremen, Bremen, Germany

* rebecca.borges@uni-bremen.de

## Abstract

Protected areas are often faced with mismatches that pose challenges to governance. Spatial mismatch is the geographical level of a conservation issue not matching the spatial level of the governance instruments for tackling this issue. It occurs in sustainable-use protected areas, for example, when local users only get involved in co-management of areas they inhabit, not of those they extract resources from. We examine the roles of key actors in transboundary communication between marine protected areas where this type of spatial mismatch was previously detected. Applying social network analysis we gauge communication networks in four Brazilian coastal-marine extractive reserves. Our results underline the importance of including qualitative data when considering network interventions for promoting integrated, cross-boundary co-management. Interviews revealed tensions between the federal, protected area Administration Agency and local users' associations. Despite high centrality, the Administration Agency is perceived as a barrier to management, while the NGO and Academia were perceived as bridges, shown through high eigenvector centralities and participants' statements. Academia appeared as an undisputed bridging partner, while the NGO was viewed with skepticism. Tensions between the users' associations, representing non-governmental actors, and the Administration Agency are caused by a lack of clarity and agreement on the roles of the government and local populations. They hinder co-management of the mangrove commons. Overall, there is a clear mismatch between the generally assigned protected area management council functioning and the actual communication and collaboration network in place. Further challenges for transboundary co-management in the study area include weak management councils, funding constraints, and digitalization issues, which point to mismatches other than just the spatial one previously identified. This

**Data availability statement:** All relevant data are located at https://zenodo.org/records/17607600.

**Funding:** This study was supported by the Rufford Foundation in the form of a grant awarded to RB (36897-2) and by the Alfred-Wegener-Institut Helmholtz Zentrum für Polar- und Meeresforschung (AWI) and the Helmholtz Institute for Functional Marine Biodiversity (HIFMB) in the form of a salary for RB and a research grant under the HIFMB Integrative Postdoc Pool (HIPP) Fellowship. RB acknowledges support by the Open Access publication fund of the Alfred-Wegener-Institut Helmholtz Zentrum für Polar- und Meeresforschung. The specific roles of this author are articulated in the 'author contributions' section. The funders had no role in the study design, data collection and analysis, decision to publish, or preparation of the manuscript.

**Competing interests:** The authors have declared that no competing interests exist.

paper highlights their impact on governance and collaboration within the extractive reserves. We propose the creation of a forum of councils to overcome these obstacles and enhance institutional collaboration. This would enhance cross-boundary communication and allow for management at a regional level that addresses transboundary challenges and alleviates various types of governance mismatches.

## 1. Introduction

Protected areas are considered critical for conserving biodiversity and maintaining ecosystem services. Managing these areas effectively requires balancing conservation goals with the needs and rights of local communities. Co-management, which involves shared governance between local communities and government agencies, has emerged as an effective strategy to address the complex challenges of managing protected areas [1,2]. Co-management is often discussed and studied alongside the concept and practice of community-based management. Community-based management typically involves local communities autonomously managing resources based on traditional practices and local knowledge [3,4]. In contrast, co-management incorporates broader stakeholder engagement, fostering collaboration between local communities, government agencies, and, on occasion, non-governmental organizations [5]. This collaborative approach aims to leverage the strengths of both local and institutional knowledge, enhancing the resilience and effectiveness of protected area management [6].

Protected areas are often faced with mismatches that pose challenges to governance. Underfunding can be considered an example of financial mismatch, while institutional mismatches involve, for instance, the inclusion of actors in management that do not cover the breath of challenges that the protected area aims to tackle [7,8]. Various mismatches can emerge together and lead to broader ones, such as spatial mismatch, where the geographical level of conservation measures does not align with the spatial scope of the issues they aim to resolve. Establishing protected area boundaries can be one of these conservation measures that lead to this mismatch, often resulting in the need for transboundary management solutions [9,10]. For instance, in the co-managed Extractive Reserves (RESEXs) along the Amazon coast, town-based management has struggled to foster cross-boundary cooperation despite the creation of a regional government administration office [11]. Persistent spatial mismatches in these RESEXs have hindered the resolution of local disputes, potentially exacerbating tensions rather than resolving them [10,11].

Using network analysis structure and dynamics of exchange and collaboration within and across protected area boundaries can be examined. Metrics such as betweenness centrality are helpful for pinpointing nodes acting as brokers, and facilitating communication and coordination between different groups [12]. Bridging actors can influence the flow of resources (including information) between other actors [13]. Such insights are critical for understanding how various stakeholders contribute to or impede effective governance [14].

Where diverse groups need to directly deliberate, network metrics alone will likely not reveal essential aspects of communication and collaboration underlining the functioning of RESEXs. Social network analysis (SNA) allows to capture the overall architecture of social interactions, quantify relationships and identify, e.g., key actors, bridging organizations or subgroups within networks [12,14].

SNA is widely applied in protected areas research to address various governance challenges, including institutional centralization, participation dynamics, and cross-boundary coordination [15–17]. SNA is used, e.g., to examine governance structures regarding communication effectiveness, power distribution, and collaboration levels [15]. It identifies changes in the centralization of institutions within a governance system [16]. In a Mexican Marine Protected Area (MPA), researchers used SNA to investigate shifts in representation and participation following regulatory interventions. This study found significant changes, including a decline in the centrality of researchers and NGO members alongside increased government engagement, which led to a more collaborative model with greater stakeholder engagement and decentralized funding. Similarly, a German nature park study utilized network analysis to gain insights into governance and communication in landscape conservation, demonstrating the existence of communicative and collaboration structures and suggesting the adaptation of the methodology for further analysis of regional and local governance structures [15].

SNA also helps analyze linkages and bridging between agencies [18] and investigate governance conditions and their implications for information flow [19]. Research in an Italian MPA demonstrated the important role of certain institutions in collecting and disseminating information, while others remained in a more peripheral context due to fewer governance interactions [19]. Beyond information flow, SNA emphasizes the importance of identifying institutional fragmentation, such as fragmentation found in protected area budgeting in the United States, which results from complex interactions between different components and stages [18].

SNA is crucial for understanding patterns of social participation in MPA management [20]. In Brazilian MPAs, a multi-layered network analysis revealed key governance issues, specifically showing low engagement of governmental organizations in management councils and/or law enforcement. Regarding cross-border challenges, complex networks analysis has been applied to evaluate the efficacy of conservation strategies for marine mammal species with transnational distributions across MPAs in Brazil, Uruguay, and Argentina [17]. The results of this cross-border analysis pointed to gaps in governance, particularly related to the lack of cross-border cooperation, with interactions limited to low-density national networks [17].

However, the nuances behind these connections, such as trust, motivations, and interpersonal dynamics are not captured. Therefore, combining quantitative and qualitative data in SNA has gained traction across various fields to provide structural and contextual insights [14].

Qualitative methods, like interviews and case studies, are crucial because social processes and interactions that drive the observed network structures can be better explored by analyzing and incorporating these case studies into SNA theory [21]. Including qualitative data is essential to uncover how informal relationships, trust dynamics, and governance challenges contribute to recognizing the importance of context. Using SNA and applying a mixed-methods approach to examining marine fisheries in Hawaii, Barnes et al. [22] show that brokerage can harm the network dynamics by acting as a constraint on individuals immersed in cultures where collective action is essential.

In an analogy with drawbridges, there are indications in the literature that actors, or groups of actors, can connect groups and promote communication and collaboration [14]. Still, they can also stop the flow, block the exchange of resources, and act as barriers to these flows, or, at least, be perceived as such barriers. Considering the importance of communication and collaboration for transboundary co-management and community-based management, a deeper understanding of how actors block and foster communication flows is essential. This could improve cross-boundary collaboration and alleviate spatial mismatches that might emerge with the boundaries of conservation initiatives, as protected areas. To our knowledge, an investigation of such "drawbridges" in cross-boundary networks of protected areas under co-management and community-based management has not yet been attempted.

We aim to fill this gap by examining the structure of communication networks between four Brazilian RESEXs in the Bragança region of Pará, North Brazil. Here we investigate the role of central actors in fostering cross-boundary communication and collaboration, acting as bridges or barriers to transboundary management. Given the importance of transboundary management in addressing mismatches in resource use, we also investigate the institutional challenges to effective management in Brazilian RESEXs. For that, we compare the roles of the local users' associations (as a representation of a community-based management approach) to those of the management councils (as a representation of a co-management approach).

Our research questions are:

1- Regarding their bridging capacity, who are the most central actors in RESEX cross-boundary connections?

2- What is the role of the RESEX management councils, and how does this role compare to that of the users' associations in the governance network of these RESEXs?

3- Does the regional interlinking of governance-relevant actor groups match the governance challenges of the four neighboring RESEXs?

This study explores strategies to overcome governance barriers and enhance collaboration by combining quantitative and qualitative methods. We show how SNA can serve as a tool to identify mismatches between spatial governance challenges and the governance network structure in place to tackle these challenges. We provide insights into how such mismatches can be avoided and coped with.

## 2. Methods

### 2.1 Study site

Our research is centered around four sustainable-use, co-managed RESEX areas in north Brazil (Fig 1). Here, local stakeholders, particularly direct users, have an active, deliberative role in the management. From west to east, these RESEXs are Tracuateua, in the city of same name (hereafter referred to as RESEX-Tr), Caeté-Taperaçu, in the town of Bragança (hereafter referred to as RESEX-Br), Araí-Peroba, in the city of Augusto Corrêa (hereafter referred to as RESEX-AC), and Gurupi-Piriá, in the town of Viseu (hereafter referred to as RESEX-Vs).

These sustainable use areas belong to the RESEX group of protected areas in Brazil's National Protected Area System [28]. According to this law, "the Extractive Reserve is an area used by traditional extractive populations, whose livelihood is based on extractivism and, additionally, on subsistence agriculture and small animal husbandry, and its basic objectives are to protect the livelihoods and culture of these populations and to ensure the sustainable use of the area's natural resources." As the definition implies, there is an emphasis on the livelihoods of the local, traditional populations. Rocha et al. [29] point out the importance of the marine RESEXs implemented on the Amazon coast to institutionalize the territory secured for traditional communities, representing an essential legal framework for maintaining their collective existence and their natural resources.

In Brazil, therefore, RESEXs are designed to protect the livelihoods of traditional communities while ensuring the sustainable use of natural resources. This model of protected area management represents a hybrid approach, combining elements of community-based management with State support [28]. Initial legislation was designed to ensure broad participation in decision-making, giving deliberation rights to the management council formed by various levels of the government and the wider society. At the same time, further legal and infra-legal instruments gave greater autonomy to local communities, which is endorsed by the political social movement that usually accompanies the creation of RESEXs [30]. Within the management councils, local communities are represented by the users' associations. This is the body that officially represents local resource users and to which territorial rights are given as a land concession act from the State to local communities when the RESEXs are created [28]. Therefore, while land concession to users' associations represents

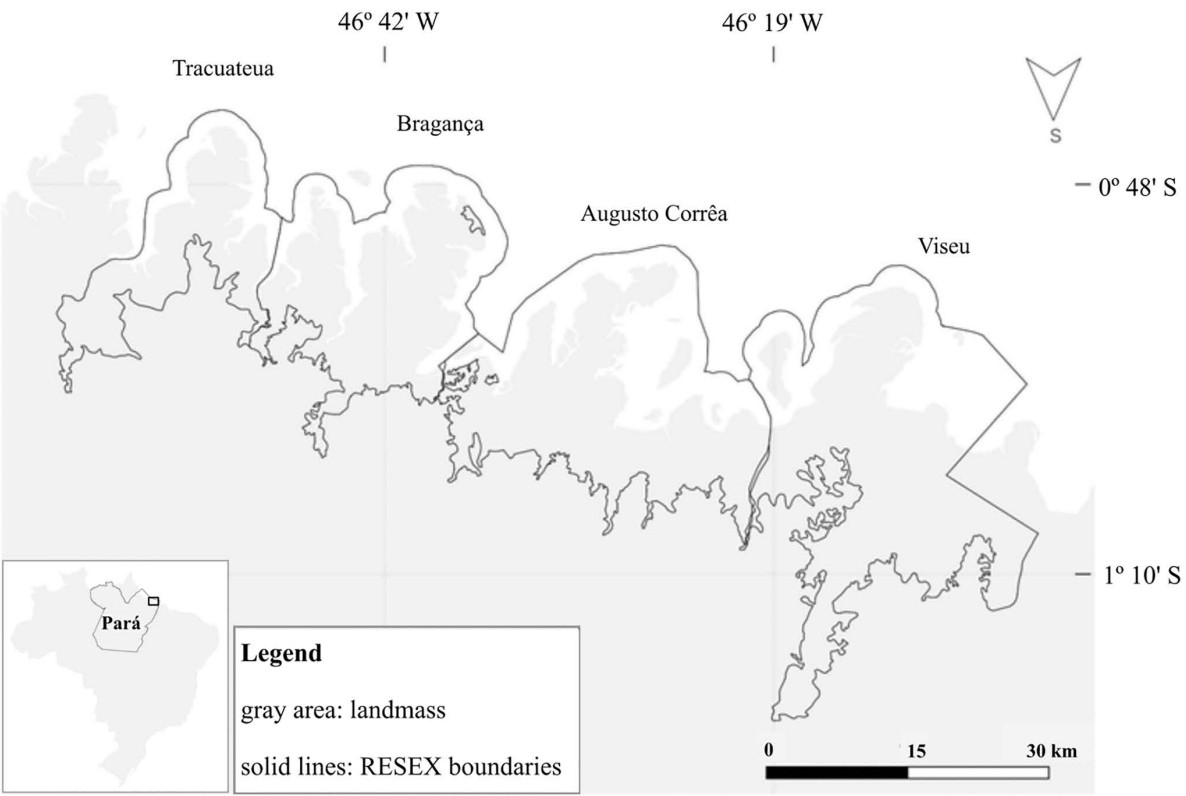

**Fig 1. Boundaries of the four neighboring extractive reserves (RESEXs) in the Bragança region of Pará, north Brazil.** Map EPSG: 4326. Data sources: ICMBio [23–26] for the boundaries kml files, and IBGE [27], for Brazil's land mass shapefile. Reprinted with adaptations from Borges [10] under a CC BY license, with permission from the journal Ecology & Society, original copyright Creative Commons Attribution 4.0 International License, 2026.

a strong community-based management element, the existence of the management councils includes other stakeholder groups in the direct management structure of these protected areas.

Each RESEX, governed by its independent management council, is allowed to develop its regulations to a certain extent. Nevertheless, it is increasingly acknowledged that local resource users from one RESEX operate in neighboring protected areas—an informal practice documented to have a decade-long history [11]. Common rules have been implemented to mitigate this spatial mismatch between RESEX boundaries and traditional fishing grounds. However, further addressing these mismatches requires extensive communication and collaboration among RESEXs and regional stakeholders, particularly in managing the cross-boundary use of fishing resources. A regional management body was created, potentially enabling excluded fishers to participate in the co-management of fishing grounds beyond their home RESEX and thereby align governance with the spatial realities of traditional fisheries [11].

Effectively managing protected areas requires communication and collaboration across boundaries, especially when ecological processes and species movements cross administrative borders, e.g., the Brazilian coastal RESEXs [31]. Cross-boundary collaboration can enhance habitat connectivity, coordinate conservation efforts, and improve the adaptive capacity of protected areas [32]. Cross-boundary governance, including communication and collaboration, is an intrinsic necessity that emerges with creating boundaries [9,10]. However, achieving such collaboration involves various social and political challenges, including aligning different management objectives and governance frameworks [33]. The Brazilian RESEXs provide a compelling case study for evaluating whether management that strongly involves local communities can simultaneously protect biodiversity and the dependent traditional communities. RESEXs face spatial mismatches that

originate from the creation of multiple protected area borders and intense cross-boundary movement. Their management councils are central to co-managing these sustainable-use areas.

Despite having a shared management or co-management perspective, the management of RESEXs has a pre-defined model, in which the state representation, in the form of the Chico Mendes Institute for Biodiversity Conservation (ICMBio), chairs the deliberative council and is also responsible for administering the funds [34].

The history of these RESEXs is rooted in social movements related to agrarian reform struggles. A study on the socio-institutional construction of marine RESEXs in Pará, including the four studied in this paper [35], observed that they were created due to the mobilization of traditional communities reacting to pressure on artisanal fishing. The importance of establishing articulated alliances between local and regional actors (universities, the Administrative Agency, and others) in the process of creating the RESEXs - which promoted events, gathering human resources and funds – has been previously highlighted [36].

These RESEXs should be managed by a deliberative or management council composed of government agencies and civil society groups:

> [Art. 18] §2 The Extractive Reserve shall be managed [in Portuguese, "gerida".] by a Deliberative Council, chaired by the body responsible for its administration [in Portuguese, "administração"] and made up of representatives of public bodies, civil society organizations, and traditional populations residing in the area, as provided for in regulations and the act that creates the protected area.

> Brasil [28], translated by the authors.

This type of RESEX ensures traditional land/sea tenure via land use concessions given to local users' associations (from here on, AUREMs - the acronym in Portuguese). Therefore, one can expect that the deliberative council should represent the interests of these populations and, thus, be formed mainly by their representatives, the most important of them being the AUREMs. The law states that the council should be presided over by the federal government agency responsible for administering all federal protected areas (ICMBio, hereafter "Administration Agency"). As seen above, the law states that this body is responsible for the *administration* of the RESEX, while the council is responsible for its *management*. This is crucial for understanding the management of these areas and identifying possible caveats and current challenges, including tension arising from potentially unclear roles of the involved groups.

## 2.2 Selected social groups for social network analysis

We interviewed 19 representatives from key groups (AUREMs, NGOs, government agencies; see Table B1 in the Supporting Information). The SNA survey was performed on the group level and used a closed actor list. We carefully prepared this list by using our own experiences from previous collaboration with groups in the study area and cooperating with a local consultant, extensively experienced working with the region's traditional populations. The list of groups was compiled based on publicly available decrees about the composition of management councils and a database constructed in 2023 by a partner project [37] comprising relevant management-related studies conducted in the region over the past 25 years, including studies such as Kasanoski [38]. The databank with the studies is available on Zenodo [39].

This study aimed to interview representatives of all the relevant groups for cross-boundary management in each RESEX. Those were either members (past or present) of groups that 1) were or had been part of, 2) had a strong interest in being part of, or 3) would at least like to have their interests considered by the management council. These groups and organizations include AUREMs, conservation NGOs, academic institutions, local, state, and federal government agencies, labor unions, community leaders, and not formally organized user groups.

After selecting the groups to be sampled, contact information of individuals who could act as group representatives was obtained. We aimed for the highest individuals in the organization hierarchy (whenever a hierarchy could be identified)

or those who worked more closely with the RESEX management. Initial contact information was obtained from publicly available sources. Missing contact information was obtained from local contacts with the prior consent of the person to be contacted.

Some of the initially selected groups no longer existed due to the extensive temporal span of the consulted documents. First contacts also concluded that some groups no longer had any interest or stake in the RESEXs. These were removed from the study before an interview had taken place. As this study focuses on regional co-management, groups exclusive to one or two of the study areas were removed from the actors list.

User groups, especially those whose activities happen close to the coast, are considered to be represented by the local RESEX AUREM, also known as "mother association". Throughout this paper, the word "regional" is used to describe groups who have a direct stake in two or more RESEXs (often the four of them) while "local" is used to refer to groups who have a direct stake in only one RESEX, with more focused action at the municipal level. The AUREMs, Agricultural Assistance, RESEX Councils, and Town Departments, despite being local groups, have counterparts in all four RESEX areas and are more likely to interact. These Agricultural Assistance bodies groups have an interest and operate in all four RESEXs simultaneously, so they are considered to be at a mixed or hybrid level. Other user-based groups, like the Rural Agricultural Association or the Artisanal Fishers' Syndicate, were not kept on the final actor list as they are local groups whose interests are usually very restricted to the RESEX where they are based. Our approach to creating the list of actors reduces the number of resource-user-based groups but still retains the most relevant groups for cross-boundary management, including the ones that were at the time most active in the councils. As each member of the council belongs to one of the organizations that make up the council, it was not possible to single out one of the members as a council representative. Therefore, no one was interviewed as a representative of the council. However, since the councils are in themselves institutions, we considered them eligible to be the target of interactions. This explains why we had 23 groups considered as part of the SNA but only 19 interviews.

Interactions with participants consisted of 1) the initial interviews, conducted in 2022 using prepared interview guidelines and including structured SNA questions; 2) follow-up questions to clarify any unclear answers or fill out gaps in our data set; and 3) focus group discussions in April and May 2024 with the directorate of the local AUREM in three of the four RESEXs studied. The recruitment period for the 2022-phase occurred between November 1st 2022 and December 12th 2022, and the recruitment period for the 2024-phase occurred from April 1st 2024 to May 10th 2024. Participants provided written consent.

## 2.3  Interviews and social network analysis survey

Semi-structured interviews (n = 19) were conducted in November and December 2022. Key stakeholders from the four RESEXs and identified governance-relevant groups were interviewed individually. Open-ended questions to obtain the qualitative data were combined with a SNA survey (see Table A1 in the Supporting Information).

The open-ended questions addressed the management of the RESEXs, including 1- how management was happening and performing during the interviews; 2- how the different groups involved communicate and/ or collaborate, and 3- if specific groups act as bridges, explicitly mentioning the Administration Agency. When RESEX management was approached, interviewees were asked if the RESEXs were managed in an integrated manner, i.e., if there was any sort of communication and collaboration in management, and what they thought could be done to improve management. (See the specific and possible follow-up questions in Table B1 in the Supporting Information.)

A stakeholder list was applied to the SNA survey. This list included the RESEX councils and consisted of 23 potential target groups, i.e., groups with which each group surveyed could interact. For each group, the person surveyed was asked if they communicated (being in meetings together, e.g.,) or collaborated (being in projects together, e.g.,) with each of the other groups. We also asked if these interactions were long-lasting or had started only recently (in the few years before the survey) and if they had changed throughout time (e.g., if they went from communication to collaboration). Further, we asked for examples of these interactions to better (re)categorize and understand them.

Here, communication is used in the sense of having any contact, as an organization, with the other group. This usually happens when groups are members of the councils, go to meetings together, and consider that the organization can reach out to the other group whenever needed, whether related to the RESEX governance or not. Collaboration is used as a synonym for cooperation or partnership. The word collaboration (or *colaboração*, in Portuguese) was chosen because it conveys a more informal sense, i.e., not limited to formal partnerships with signed agreements.

Even if not explicitly prompted, surveyed individuals often explained their relationships in detail. For instance, mentioning why a given interaction had ceased, if an interaction had decreased or increased in intensity, or if and why interactions had gone from more active collaboration to merely being in meetings together. Additionally, respondents talked about the COVID-19 pandemic and recent political developments in Brazil and their effects on overall social welfare and on the RESEX management.

Interviewed persons were asked to answer questions for the group and not to consider their private interactions. However, we acknowledge the caveats of interviewing an individual as a representative of a group, understanding that in whichever aspect of an organization interpersonal relationships play a role. The lines that separate individuals from the group of which they are part are blurry, and a strict separation cannot be achieved.

**Qualitative analysis.** The responses to the qualitative questions were audio recorded, transcribed and analyzed using MAXQDA 2022 [40]. A simple text document was created for the coding process by listening to the interviews and taking notes of any relevant information mentioned by the interviewees. This included keywords or terms mentioned and parts of quotes that were further obtained from the transcripts. Transcripts were consulted for clarification purposes. Any quotes in this article are translations from the original, in Portuguese, performed by the authors.

Inductive-deductive coding was used for the initial note-taking, in which themes are developed directly from a close reading of the transcript of the interviews. This coding process can be defined as inductive-deductive because, even though the initial coding of the material was done using an inductive approach, i.e., without predefined codes, the general topics of the interviews were driven by the specific questions asked, following a merged approach [41]. A first set of codes was created through identification while reading the documents, including perceptions, opinions, and feelings around the RESEXs that might relate to the topic of this study.

The resulting codes were based on the following topics: a- role of each group in the survey, tensions between groups, examples of good collaboration, current management of the RESEXs; b- reasons for good/bad relationships; and c- reasons for good/poor functioning of a particular group, with an emphasis on the RESEX councils.

The qualitative results are illustrated by quotes from the interviews. Unless otherwise mentioned, the quotes are from the individual interviews, following the topics detailed in Table A1 in the Supporting Information. While the most representative quotes are kept in the main text of this manuscript, additional interview excerpts that illustrate the qualitative findings are provided in the Supporting Information (SI Section F) and are cited throughout the results in the format (SI FX.X).

**Quantitative analysis.** SNA data were tabulated via commonly used digital spreadsheets. The data were later inserted into ©Gephi 0.10.1 [42], an open-source software tool for visualizing and analyzing networks, allowing for the customization of networks and maximizing the visual rendering, which can help convey results through network images.

**Network visualization.** In network analysis, each network component capable of interacting with others is called "node" or "vertex". The interactions, or the connections between the nodes, are called "links" or "edges". Nodes and links are displayed in a network using different rules or settings. In Gephi, packages of rules are called "layouts."

For this study, we used the layouts ForceAtlas2 and LabelAdjust. ForceAtlas2 is a force-directed algorithm that can better show clusters than Fruchterman-Reingold [43]. The chosen parameters for each layout are provided in Supporting Information C.

**Link direction and weights.** Our methodological approach generated information about the direction of the reported interactions. This means there is a distinction between communication signaled by Group A toward Group B and

communication signaled by Group B toward Group A. It is possible that Group A, for example, reports communication with Group B, while Group B reports not communicating with Group A. The analysis considers, therefore, the interactions' directionality (see below).

Weights were attributed to the links according to their type, namely communication and/ or collaboration. Any link got one point if one of these interactions was reported or two points if both types of interaction were mentioned. For each long-lasting interaction, an additional point was attributed to the interaction. Therefore, the weights range from 0 (no inter-action) to 4 (communication and collaboration, both long-lasting).

Some respondents did not reply to the survey questions regarding the past (i.e., if the interaction was long-lasting or had only recently been established), mainly because they were not present during this time and did not have the respec-tive information. Thereby, 32 interactions of past communication and 33 of past collaboration could not be confirmed. This impacts the weight attribution, which, for this reason, gives higher weight to the interactions maintained by organizations where the respondent has been there for a long time. However, this aspect is approached in the weight vs. non-weighted degree analysis, in which the influence of the weights is gauged.

**Analyzed network parameters.** Using Gephi, we analyzed the following parameters for each node:

a- **Betweenness centrality**: directed. This metric quantifies the number of times a node serves as a conduit along the shortest path between two other nodes [12]. Betweenness centrality is crucial for identifying nodes that can facilitate or inhibit communication between disparate network parts. The default shortest path betweenness centrality algorithm provided in Gephi was employed for this analysis, with normalized values.

b- **Eigenvector centrality** (0–1): directed, with 100 iterations. According to Gephi, eigenvector centrality measures the importance of a node in the network based on its connections. Eigenvector centrality was employed in this study to assess the influence of nodes based on the importance of their neighbors [44]. Nodes with high eigenvector centrality frequently connect to other influential nodes, which may indicate central connectors within the network.

Eigenvector vs. Betweenness Centrality: Eigenvector centrality and betweenness centrality provide distinct insights into the roles of nodes within a network. Eigenvector centrality measures a node's influence based on the centrality of its neighbors, giving higher scores to nodes connected to other influential nodes [44]. In contrast, betweenness centrality identifies nodes that serve as critical points of connection between different parts of the network, highlighting nodes that frequently act as bridges in the shortest paths between others [44].

c- **Weighted in-degree**: In-degree represents the number of incoming links to the node. Weighted degree calculates the sum of the weights of all links a node has. Therefore, a high weight degree can indicate many connections with average or low weights or few connections with high weights. Very high values usually relate to nodes with many highly weighted connections, while very low values are associated with nodes with only few low-weighted connec-tions. The quality or weight of links is considered, offering a deeper understanding of the node's practical influence in the network [45]. In this study, the weights relate to the duration of the interactions and whether these also include collaboration, i.e., joint projects and other cooperative initiatives. We aim to understand the role of interactions beyond the present-day exchange of information. In-degree is used because the RESEX councils only display incoming interactions, as no council representative could be interviewed. Interviewees confirmed the inactivity of the RESEX councils. Therefore, in our study, in-degree serves as a measure comparable for all involved actors and we are confident that our findings give meaningful insights into the network dynamics of the RESEX co-management system in the Bragança region.

To gauge the role of the RESEX councils and AUREMS, we remove them from the networks and see how that impacts the network measures of the remaining nodes (Table 1).

**Table 1. Network measures of organizations involved in the governance of four RESEXs, on the coast of Pará, north Brazil. The highest five values per measure are highlighted by gray shades and thick box borders.**

| organization | whole network | | | without RESEX Councils | | | without RESEX councils and AUREMS | | |
|---|---|---|---|---|---|---|---|---|---|
| | betweenness centrality | eigenvector-centrality | weighted in-degree | betweenness centrality | eigenvector-centrality | weighted in-degree | betweenness centrality | eigenvector-centrality | weighted in-degree |
| **regional** | | | | | | | | | |
| Province Department | 21.915 | 0.689 | 40 | 14.3274 | 0.689 | 40 | 8.3 | 0.689 | 38 |
| Monitoring Agency | 12.045 | 0.469 | 20 | 10.151 | 0.469 | 20 | 6.246 | 0.469 | 13 |
| RESEX Confederation | 6.1397 | 0.522 | 17 | 5.302 | 0.522 | 17 | 1.5 | 0.522 | 11 |
| Administration Agency | 34.719 | 1 | 54 | 26.655 | 1 | 54 | 19.2095 | 1 | 44 |
| Academia | 32.157 | 0.956 | 49 | 25.260 | 0.956 | 49 | 17.585 | 0.956 | 40 |
| Navy | 0.536 | 0.370 | 18 | 0.536 | 0.370 | 18 | 0.6 | 0.370 | 14 |
| NGO | 8.992 | 0.754 | 23 | 5.612 | 0.754 | 23 | 1.7 | 0.754 | 15 |

## 2.4 Focus groups

In April and May 2024, initial interviews and SNA survey results were presented to the council members of three of the four AUREMs. The reception of the results by local users is also considered in this study. The AUREM presidents, representatives of the village hubs, and other guests invited by the AUREMs participated in our focus group discussions. The exact composition of the groups varied and will not be detailed here to guarantee the anonymity of the attendees. The vast majority of the attendees had not been surveyed in the first phase of our study. Specific details regarding the composition and attendance numbers for the three focus groups are provided in Supporting Information H.

The data collected while conducting the focus groups were analyzed, as was the interview data. However, instead of specific questions, the focus groups were presented with and asked for their opinions on the results from:

• an earlier study, which showed cross-boundary movements of crab fishers [11], followed by

• some of the interview questions shown in Supporting Information A, especially regarding the need for integrated management and the ideas to achieve that, e.g., the creation of a forum to bring together representatives of the management council from the four RESEXs; and finally

• the SNA survey described above, mainly the visualization of the networks, and the general interpretation of centrality and brokerage. This initial interpretation is provided in Supporting Information D.

## 2.5 Legal documents

The data for the analyses in this paper come mainly from interviews, surveys, and focus groups with local actors. Additionally, these results were complemented by legal documents such as legislation and meeting minutes that directly relate to the governance of the investigated RESEXs. Available public minutes of council meetings, obtained via local university partners and the online registry for research projects in federal protected areas [46], were also consulted.

## 2.6 Ethics approvals and research permits

In Germany, ethics approval was obtained from the University of Oldenburg (Drs.Nr.EK/2022/053). Additional research permission was obtained to perform research inside the RESEXs from the *Sistema de Autorização e Informação em Biodiversidade* – Sisbio, the Brazilian Biodiversity Authorization and Information System (82744−1).

## 3. Results

### 3.1. Centrality, bridging, and influential capacity

At first glance, the network resulting from our data is interconnected, with no clear subgroups or outliers in the network structure (Fig 2). It contains 12 nodes from government bodies and 11 nodes from civil society, academia, and the NGO.

The regional groups tend to have a more central place in the network, compared to the local groups. Four organizations stand out in the overall network. These four organizations are among the five organizations with the highest values for betweenness, eigenvector, and weighted in-degree centrality, namely the Province Department, the Administration Agency, Academia, and the AUREM in RESEX-Tr. By reaching high values in all three measures examined in this study, they have a high bridging capacity, are influential due to many connections with central influential actors, and have the most and/ or many long-lasting connections. Table 1 provides an abridged overview of the network measures, focusing on the key regional organizations and the highest-ranked RESEX-specific organizations. The complete quantitative dataset, including the full list of all 23 organizations and the specific RESEX-based values for each Town Department, Agricultural Assistance body, and AUREM, is provided in Supporting Information I.

The Monitoring Agency is among the five nodes with the highest betweenness centrality values. The NGO displays a high eigenvector centrality, maintaining many connections to influential central nodes. The AUREM in RESEX-Vs is among the five nodes with the highest weighted in-degree, showing that this AUREM has many, or at least considerably more, long-lasting connections than most of the other involved organizations.

Comparing the positioning of groups present in each of the RESEX (Agricultural Assistance, Town Department, AUREM, Management Council) in the networks shown in Fig 2, the Agricultural Assistance bodies, Town Departments, and AUREMs form clear clusters. At the same time, the Management Councils are spread out. This finding is more deeply discussed in section 3.2.

All regional groups seem to form clusters at the periphery of the networks, while the RESEX councils have more central positions. Three RESEX councils (Br, Tr and Vs) are connected to all other group representatives from their respective RESEX, namely the Agricultural Assistance bodies, the AUREMs, and the Town Departments. In contrast, the Management Council in RESEX-AC has only one incoming link from within this same town (AC). This link departs from the AUREM in RESEX-AC, meaning that only the AUREM (the users' association) claimed to be connected to the Management Council in RESEX-AC.

Removing the RESEX councils only slightly decreases the betweenness centrality values of all other nodes in the network. All other network measures remain the same whether the councils are in- or excluded from the network. The councils seem not to influence the network structure/ positioning of the other nodes, because they can be removed from the network without effect.

The removal of councils and AUREMs increased the importance of the Town Departments and the Agricultural Assistance bodies. They are now among the five nodes with the highest values for betweenness centrality (Town Department Vs), eigenvector centrality (Town Department Tr), and weighted in-degree (Agricultural Assistance AC, Town Department Br, Agricultural Assistance Br, Agricultural Assistance Tr, Town Department Vs, Agricultural Assistance Vs). The ranking and indicated roles of the organizations within the network remain the same.

Six regional groups, namely the Administration Agency, the Monitoring Agency, Academia, the RESEX Confederation, the NGO, and the Province Department, as well as two local actors, the AUREM of RESEX-AC and the AUREM of RESEX-Br, connect the RESEX Councils. All eight of these bridging actors address all four of the RESEX councils. They are among the top five with the highest values of at least one measure in the overall network (Fig 2). These actors are therefore very suitable as connecting points to the overall network. Visualization and network measures of the network, focusing on the RESEX Councils and their bridging nodes, can be found in Supporting Information.

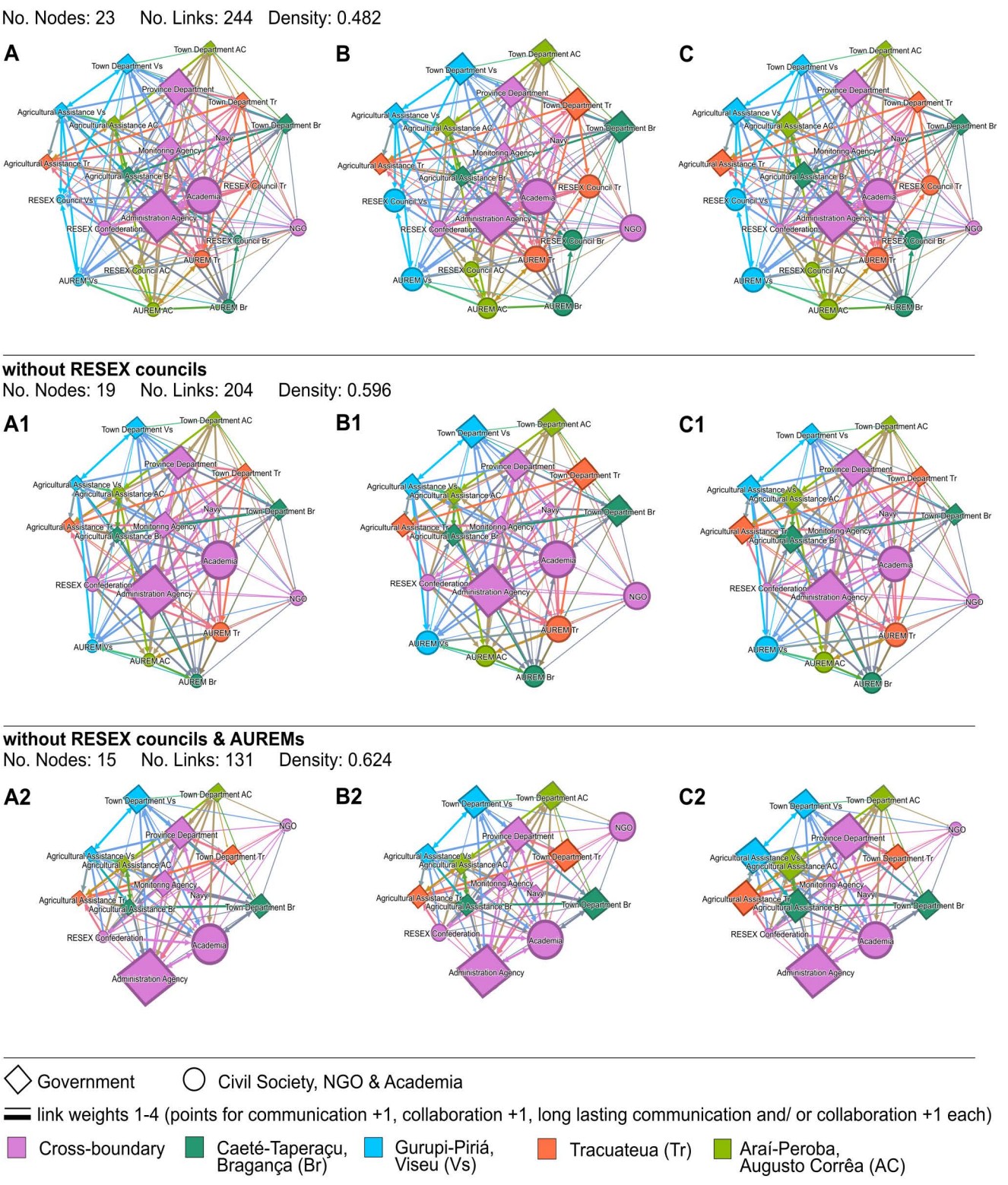

No. Nodes: 23    No. Links: 244    Density: 0.482

**without RESEX councils**
No. Nodes: 19    No. Links: 204    Density: 0.596

**without RESEX councils & AUREMs**
No. Nodes: 15    No. Links: 131    Density: 0.624

◇ Government          ○ Civil Society, NGO & Academia

⬛ link weights 1-4 (points for communication +1, collaboration +1, long lasting communication and/ or collaboration +1 each)

🟪 Cross-boundary    🟩 Caeté-Taperaçu, Bragança (Br)    🟦 Gurupi-Piriá, Viseu (Vs)    🟧 Tracuateua (Tr)    🟩 Araí-Peroba, Augusto Corrêa (AC)

**Fig 2. Cross-boundary co-management network of four extractive reserves (RESEXs) in the Bragança region, Pará, north Brazil.** Node sizes represent A Betweenness centrality (Range 0 - 34.719), B Eigenvector centrality (Range 0.37 - 1), and C weighted in-degree (Range 17 - 54). Edge color results from the color of the sending and the receiving node. Therefore, mixed edge colors indicate connections between nodes from different administrative areas.

## 3.2.  Regional and local link distribution

With 29 out of 244 links, the regional organizations share the most considerable proportion of links within the whole network with each other (Fig 3). They maintain 22 links towards organizations from each RESEX. The regional organizations address the local organizations with 27 links in total (links to all).

With nine links each, organizations in RESEX-Tr and RESEX-Vs are more interconnected within their RESEXs when compared to Br (8) and AC (5), i.e., they are better connected to spatially closer groups. All local organizations maintain more links to regional organizations (13; 14) than they do with organizations of other RESEXs.

The smallest number of links is shared directly between organizations from different RESEXs (3–5). RESEX-Br and RESEX-AC maintain slightly more links towards organizations from other RESEXs (4; 5) than RESEX-Tr and RESEX-AC do (3). The highest number of connections are either within a given RESEX or towards regional groups, i.e., the RESEXs do not connect well among each other.

## 3.3  Perceived challenges for cross-boundary and integrated management

The qualitative data examination also provides information on how budget constraints and digitalization issues affect the shape of these local networks, impacting collaboration and governance of the four RESEXs.

The interviews show that the creation of the Administration Agency's integrated management office (NGI) has not been sufficient to promote integrated management despite joint administrative tasks. An Administration Agency staff member identified a joint forum as a potential solution but immediately recognized the constraints: securing resources to host people, provide meals, and facilitate travel remains a fundamental barrier. At the time of the interviews and focus groups (2022–2024), informal coordination happened through a WhatsApp group of AUREM presidents, with occasional in-person meetings—sometimes requiring the staff member to personally fund participants' transportation (SI F1.2).

These funding constraints created tensions between AUREMs and the Administration Agency. AUREM representatives expressed frustration about the partnership imbalance, questioning the value of a partnership where they lack resources for basic needs like transportation while the Administration Agency has vehicles and fuel but limited presence in the associations' operations (SI F1.2).

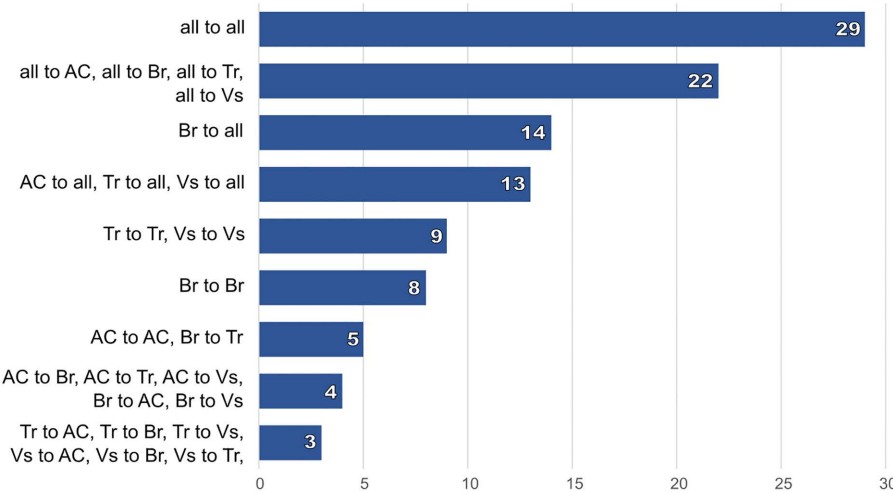

**Fig 3.  Count of links within and between administrative regions (RESEXs) along the coast of Pará, North Brazil. all = regional organizations, AC = RESEX-AC (Araí-Peroba, Augusto Corrêa), Br = RESEX-Br (Caeté-Taperaçu, Bragança), Tr = RESEX-Tr (Tracuateua), Vs = RESEX-Vs (Gurupi-Piriá, Viseu).** Total count of links in the network: 244.

Beyond funding, restricted digital skills and inadequate internet infrastructure limit closer collaboration between the four RESEXs. While the COVID-19 pandemic demonstrated the potential for remote collaboration, implementation faced multiple barriers: poor internet connections, lack of appropriate devices, limited digital literacy among some leaders, and data costs for extended online meetings. As one AUREM president acknowledged, learning to facilitate online meetings remained a work in progress, though there was willingness to develop these skills (SI F2.1).

Furthermore, the years immediately before the interviews had been difficult for RESEX management due to the COVID-19 pandemic (2020–2022) and massive political changes beginning in 2016. A representative of the NGO reported these as causes for the inactive councils, noting that the Administration Agency spent 2021 primarily distributing food parcels, with council agendas remaining stalled since approximately 2019. This demobilization resulted from multiple factors: the pandemic, federal-level dismantling and resource cuts, and reduced staffing at the Administration Agency (SI F1.3).

Activities were slowly resuming in 2022 and continued through May 2024 when focus groups took place, including concrete government measures such as releasing funds for subsidies.

## 3.4  Roles of stakeholder groups

### 3.4.1  The role of the Administration Agency.
In the interview, the Administration Agency claimed to maintain good relationships with all the main actors in the management processes, highlighting academic partnerships, but also reported time restrictions for their presence and activities (SI F3.1).

Both local users and other government bodies come to the Administration Agency for RESEX management-related issues and permits. This is perceived as pressure but also recognized as a matter of responsibility and power. This role is evident through the increased demand from beneficiaries and municipal entities who consistently approach the Administration Agency for necessary permits related to fishing traps or construction, recognizing its practical management authority (SI F3.2). Furthermore, the Administration Agency is described as the *de facto* body managing the RESEX, instead of the RESEX councils (SI F3.3).

Some actors view the Administration Agency's role as disenfranchising the other groups. Local users criticize the Administration Agency's central role, claiming that there is an attempt to populate the councils with government-related groups, such as the municipal Department of Environment and the Administration Agency itself.

From the users' perspective, represented through the AUREMs, the Administration Agency also seems to "stand in the way" between RESEX councils, instead of using their network positioning to facilitate communication and collaboration (see Section 3.1 and Supporting Information C).

### 3.4.2  The role of the AUREMs.
The NGO, which works mainly supporting the AUREMs, acknowledges their key role in the RESEX management:

The associations [AUREMs] themselves [...] are a very fundamental part, very key in the deliberative council [...] They play a very important role as representatives of the communities and as mobilizers of community people as well.

(NGO staff)

However, some actor groups point out several deficiencies of the AUREMs, among those, missing representative status (SI F4.1 and 4.2). For instance, one Town Department representative stated that AUREMs can become a "problem in society" when board members lack the management capacity necessary to handle administrative processes (SI F4.1). Furthermore, Agricultural Assistance staff expressed concern that AUREMs are often centered on just one person, diminishing the necessary participatory nature expected of these groups (SI F4.2)

The criticisms emerge from acknowledging, like the NGO does, the importance of the AUREMs and the expectations regarding their functioning as main community-based managers of the RESEXs. They point to a common desire to see

improvements in the functioning of the AUREMs, as opposed to a change in the role that these groups are expected to play in the overall RESEX governance.

**3.4.3 Tensions between the Administration Agency and the AUREMs.** In an individual interview, the president of one of the AUREMs hinted at a tension, or emerging conflict, between the AUREMs and the Administration Agency:

> [The Administration Agency] is the government and one part that helps with management. The other part is us [AUREMs] who have the concession. The real right to use this territory. It is not [the Administration Agency] that has it. It's us. We know where the shoe pinches [...] Decision-making is here.

> (President of one of the AUREMs)

When members of this same AUREM were presented with the visual results at the focus groups, they interpreted that the Administration Agency, exercising its role as administrator of the RESEXs, would be trying to take over the central role of the local populations. The AUREMs claim a more central role. They fear that the council is being populated by government bodies, as mentioned during an individual interview. They seem to agree with the view that government-related agencies are trying to take over the seats in the councils, reducing the role and power of the local users and the groups that represent them (SI F5.1)

Perceptions captured during the presentation of the first network maps to the AUREMs during the focus group discussions confirmed the underlying tension that is not distinguishable by the researchers' visual or statistical analysis of the network. In terms of methods, therefore, we see how qualitative data, from both the individual and the focus groups, can play a key role in revealing hidden characteristics of the network, especially regarding the main government body's role in the governance of these RESEXs.

There is a lack of trust and fear concerning the approaches of various levels of the government to undermine the representation of the local users, which can, at the moment, not be verified or measured. At the time of writing, the councils are still being formed. A legal instrument encourages a balanced distribution of seats: "§3 The representation of public bodies and civil society on the councils should, whenever possible, be equal, taking into account regional peculiarities." [47]. However, this is not a strict regulation with a fixed ratio of seats between government and non-government groups. Additionally, in paragraph §3 civil society is considered counteracting the public bodies. Therefore, the mentioned equal share of seats does not assign half of the seats purely to the local users; it includes the scientific community, NGOs, and private entrepreneurs, as also specified by this instrument in Paragraph §2.

The AUREMs and the Administration Agency agree that the council is the decision-making platform. However, the question of where the management takes place and who is the best informed and acting body in terms of RESEX management and the everyday functioning of the RESEX is highly controversial:

> The managing body is not the council. The managing body is the Administration Agency. The council is a decision-making body and so on, but the actual managing body, which is the one that "fixes the car", is the Administration Agency, not the council.

> (Administration Agency staff)

There is no clear distinction between what administrative and management tasks are. AUREMs express the need to be more present and take a more critical role in many RESEX-related tasks that the Administration Agency perceives as administrative.

Social research conducted in RESEXs, for example, need ethical approval by the ethics committees in the Brazilian universities, and an additional permit is granted via a federal system - Sisbio (for federal protected areas), in which the research proposals are analyzed by the RESEX manager (Administration Agency staff). This process does not involve

the deliberative councils of the RESEX on the ground. The Administration Agency's RESEX manager decides whether a proposal will be discussed in meetings with the AUREMs or the RESEX council. One of the presidents stated that the AUREMs would like more control over the permit granting and that an extra approval, with a specific agreement, should happen via the AUREMs.This aspiration was addressed by a representative of the Administration Agency and clearly described as operatively unrealistic. They defended the operative choice of centralizing research permit concession, arguing that if all research proposals were brought before the council, meeting agendas would be overwhelmed, exceeding the scope of the limited council meetings (SI F3.4)

This position is understandable, noticing 1) the scarcity of the respective RESEX council meetings (according to informants, for some of the RESEXs approached here, nearly no council meetings had been held during the pandemic years) and 2) the high volume of permit requests sent via Sisbio. Sisbio data (SI G1) show that 66–174 permits were granted per RESEX between 2007 and 2024, corresponding to an average of approximately four to ten projects/year. Furthermore, minutes show that the annual frequency of ordinary council meetings varies from zero to five. This calculation supports the Administration Agency's argument that two permit requests might need to be discussed per meeting, making the AUREMs' claim for control time-wise difficult.

Other participants would prefer a system where the AUREMs are responsible for bringing actors together under the banner of co-management and the Administration Agency would take a more managerial role, by administering everyday tasks in the RESEXs, that can be done routinely and do not require long debating and reflecting:

[...] we [the AUREMs] do the co-management [*cogestão*], and they [the Administration Agency] do the management [*gestão*].

(President of one of the AUREMs)

In RESEX-Tr, the management is geenrally acknowledged by interviewees as being more balanced, i.e., the AUREM has a better relationship with the Administration Agency and can better coordinate their activities and roles in the RESEX management. A local Town Department representative noted that, in RESEX-Tr, management appears balanced, with the Administration Agency handling administrative tasks while the AUREM maintains an organizational role (SI F3.5). This is in line with the very central position AUREM-Tr has within the overall network, as discussed above.

In contrast to the current managing situation, the intended centrality of local communities is echoed when participants state what the essence of the RESEX means for them:

The RESEX area is a protected area where users have priority, and it has to be sustainable.

(Agricultural Assistance staff)

This position encounters resonance in the legal definition of the RESEX and illustrates the mismatch in governance, which shifts management centrality towards actor groups other than the ones led by local community members.

**3.4.4 The role of the NGO and Academia.** Contrary to the roles of the Administration Agency, the roles of the NGO and Academia seem to be less controversial. The NGO's role has been largely seen as positive, with municipal administration staff noting that the NGO helped unify the RESEXs through initiatives like the COASTAL500 project, which improved information sharing and connectivity between different reserves (SI F6.1)

Although the projects led by the NGO are recognized as necessary, some skepticism arose concerning this relatively new NGO among the AUREMs, the Administration Agency and Academia. The NGO was perceived as trying to impose its agenda. This changed by an open dialogue, growing a partnership with shared activities and mutual support (SI F6.2)

As for the role of Academia, the researcher interviewed pointed to the central role played by the university, and the network analysis results confirmed that. While most interviewees claim that their group has a good relationship with

Academia, they also recognize the need for increased return of the results of research done in the area. While acknowledging that professors and researchers "are playing their part" and that research is vital for the survival of the fishermen, a member of one AUREM stressed the importance of the university "put[ting] what we want to learn back into our community, into the society"(SI F7.1)

Despite this recognized importance, discontent was expressed by one AUREM president who insisted that they welcome research only "as long as the association [AUREM]... and the people in the community agree," noting that their partnership with the local university was "kind of scratched" due to the lack of research return (SI F5.2), both in terms of divulging the results of the research but also in applying this research to the management of the RESEX and ultimately, an improvement of the quality of the lives in these local communities.

## 4. Discussion

### 4.1. Bridging communication and collaboration for cross-boundary protected area management

The network analyses identified key groups in the four RESEXs that function as bridges for cross-boundary protected area management. The Province Department, Administration Agency, Academia, and AUREM-Tr hold the most central positions in the network, facilitating information exchange and collaborative initiatives. As organizations with numerous long-lasting connections, they are unlikely to lose influence within the governance network.

The central role of AUREM-Tr stands out as it is a local organization specific to one RESEX, unlike the other bridging actors, which are regional groups. RESEX-Tr has successfully acquired financial resources (e.g., EU funding) [11], mainly due to the exceptionally active role of its president, as evidenced in our interviews. AUREM-AC, on the other hand, only became effective in 2017, to reactivate the council [48]. This contrast shows that AUREMs are active to varying degrees in these RESEXs.

While NGOs and Academia function as facilitating or bridging groups in the network (supported by Academia's central position and the NGO's high eigenvector centrality), the NGO's facilitator role is viewed with reservation, particularly regarding its early efforts to establish communication channels.

Local organizations play minor roles in cross-boundary communication. Only when regional civil society organizations, AUREMs, and RESEX Councils are removed, some local organizations gain importance. Meanwhile, regional organizations like the Province Department, the Administration Agency, and Academia maintain their central roles, demonstrating their well-established network positions.

### 4.2. Governance mismatches and barriers to effective co-management

**4.2.1 Spatial and other mismatches in RESEX governance.** Our analysis revealed the most significant governance challenge: the misalignment between RESEX boundaries—established according to town limits—and traditional fishing grounds, creating what conservation scholars call a spatial or scale mismatch. As Cumming et al. [49] explain, such mismatches occur when management's geographical scale does not align with the ecological problem's scale. Fishers can only participate in co-managing fishing grounds within their RESEX, while cross-boundary fisheries are subject to management that largely excludes traditional users [11].

This spatial mismatch is compounded by a functional mismatch in the councils' role. Surprisingly, the RESEX councils are not central actors in our network analysis, and their exclusion barely changes the network structure. This indicates their minor role in collaboration and communication—a stark contradiction to their intended function as decision-making platforms according to legislation [28]. Instead of civil society actors managing RESEXs through councils, most influence is exerted by governmental actors, Academia, and NGOs.

The councils appear scattered and disconnected in the governance network. Unlike other groups (Agricultural Assistance offices, Town Departments, and AUREMs), which cluster and connect well among themselves, there is no direct

connection between RESEX councils. This can result in slow communication, limited information exchange, and a lack of collaboration—an apparent contradiction between the assigned council function and the communication network.

Inter-RESEX interaction among users typically occurs via the AUREMs, which connect through the RESEX Confederation or NGO activities. Direct cross-boundary connections between local organizations are rare; instead, they maintain connections to regional organizations interested in all four RESEXs. This structure gives regional organizations the power to facilitate or block cross-boundary activities.

AUREMs have maintained cohesion despite being distanced from government-related groups. Supported by the NGO, AUREMs have continued their work even during budget cuts. Thus, they have assumed roles that councils struggle to maintain, potentially creating conflict between government entities and users due to the absence of a unified co-management platform.

**4.2.2 Institutional role mismatches and trust deficits.** As previously mentioned, the RESEX councils are not among the central actors in our analysis. Their exclusion does not generally change the network structure – they currently play a minor role in the examined collaboration and communication network. According to our network analysis and the qualitative interviews, the RESEX councils currently have no capacity to act as communication bridges, as they are supposed to. Instead of civil society actors, especially local community groups, which send representatives to form the RESEX council and manage the respective RESEX, most influence within the communication and collaboration network is expressed by governmental actors, Academia and the NGO. This goes against what is expected from RESEX management, considering the role that the RESEX councils, as decision-making platforms, should have in these RESEXs, according to the pertinent legislation [28].

A recent study in the region already showed the councils' deficiencies. Examining the councils in the RESEXs Br and AC, it showed that, although territorial governance is a social innovation for managing marine RESEX territories, RESEX councils lack effective governance. The study identified weakened institutional arrangements and ineffective participation of extractive communities in decision-making [48].

The unclear delineation of responsibilities between different actors represents another critical mismatch in RESEX governance, which one could name an institutional mismatch. While the Administration Agency and Academia occupy central network positions with potential to act as bridges, AUREM focus groups view the Administration Agency as a barrier rather than a bridge. This perception stems from ambiguity in governance roles and the associated lack of trust.

The legislation (Brasil [28] Art. 18, §2) clearly states that a RESEX "shall be managed by a deliberative council" where the Administration Agency is one component alongside traditional populations and civil society. However, the Administration Agency is often perceived as attempting to increase its influence and power within the RESEX.

The tension regarding the granting process of research permits exemplifies this institutional role mismatch. The Administration Agency considers permit approval an administrative task since scientific research is integral to these RESEXs. Conversely, AUREMs seek more transparency and view research as a policy-related management aspect that should be discussed in council meetings.

Well-defined institutional roles are crucial for effective common-pool resource governance, providing clarity, accountability, and facilitating collective action. Without clearly defined roles, polycentric systems become dysfunctional, leading to responsibility overlap or enforcement gaps [50]. While roles should be clear, they must remain adaptable to changing conditions, as rigid institutions typically fail in long-term resource management [51].

The study region's history of extensive research has generated trust issues. Residents often complain that research "never comes back," with expectations ranging from researchers presenting results to concrete quality-of-life improvements. Research that bypasses proper permit procedures also contributes to mistrust, especially among AUREMs. Furthermore, a lack of trust across groups makes brokers in the networks significantly less productive in cases where "social identities are strong and divides between groups pronounced" (Barnes et al. [22], p. 64). This applies to the social divide between the government and local users in the study area.

 

**4.2.3 Resource mismatches and capacity limitations.** Significant resource mismatches further constrain the effectiveness of RESEX governance. Co-management success depends on local actors' active involvement, but their role is weakened by insufficient investment in capacity. Agrawal and Ribot [52] and Ribot [53] argue that decentralization fails when local populations do not gain control over necessary governance resources, including financial support and capacity building.

Two decades ago, before the creation of the RESEXs, capacity building for local populations, who hold rights to use these territories through AUREMs, was already considered essential for local management [54]. However, the Administration Agency, which centralizes administrative tasks, has limited financial means to invest in local capacity building. In recent years, external actors, like the NGO, have stepped in to strengthen AUREMs and implement capacity-building programs.

RESEXs face substantial obstacles due to insufficient funding for councils. Council members need concrete support to attend meetings and engage politically, particularly fishers, who lose a day's income when attending meetings. Financial shortfalls also contribute to resistance to digitalization due to a lack of infrastructure—councils lack laptops, computers, and reliable internet connections necessary for online meetings.

Creating a platform or forum to integrate RESEXs and their councils requires funding, which depends on third-party projects since government funding is insufficient. This resource scarcity also constrains what councils can realistically operate and how workload can be distributed.

The COVID-19 pandemic further eroded councils' influence, as digitalization was not widely adopted, particularly among older generation leaders unfamiliar with technology. Despite persistent governance challenges, post-pandemic management has reverted to in-person initiatives but has not recovered the mobilization capacities that led to the creation of these protected areas.

Building social capital through capacity-building initiatives is essential for improving local resource management where financial and infrastructural constraints exist [55]. Strengthened management councils could mitigate tensions and improve cross-boundary integration. Incorporating multiple knowledge systems and fostering cooperation in co-management arrangements to build resilient, adaptive governance structures is key to strengthening these governance arrangements [2,56] such as the councils.

RESEXs represent a confluence of synergies and tensions arising from a governance framework combining co-management and community-based strategies during socioeconomic crises marked by strong dependency on federal administration and financial support. The ambitious goal of bringing diverse groups into management councils, expected to deliberate on RESEX functioning, faces multiple mismatch challenges that must be addressed to achieve effective governance.

## 5. Conclusions

Our mixed-methods approach demonstrates the value of combining qualitative data with network analysis to understand complex governance structures. Qualitative data effectively contextualized network results and provided critical feedback on preliminary findings, particularly evident when local users' responses altered our interpretation of management groups' roles in these networks. The results show that the governance of the four neighboring RESEXs is characterized by interconnected mismatches that hinder effective cross-boundary management.

### 5.1. Mismatch 1: Spatial and functional alignment

The governance structure is marked by two key alignment issues: First, a spatial mismatch exists between RESEX boundaries (established according to town limits) and traditional fishing grounds. Second, a functional mismatch is evident in the RESEX councils' actual role, which is relegated compared to their intended legislative function as decision-making platforms.

The RESEX councils—legally designated primary management bodies—appear scattered and play minor roles in bridging and bonding actor groups within the network. Their exclusion barely changes the network structure, indicating their diminished role, which is perhaps the most critical mismatch in the governance structure. This lack of direct connection between RESEX councils results in slow communication, limited information exchange, and a lack of collaboration.

### 5.2. Mismatch 2: Institutional roles and trust deficits

Institutional role mismatches create tensions between government bodies and local populations due to unclear delineation of responsibilities. Although the legislation states that the council manages the reserve, and the Administration Agency performs administrative tasks, the Administration Agency is often viewed as the *de facto* managing body.

Meanwhile, regional organizations including the Province Department, Administration Agency, and Academia maintain well-established central positions. While NGO and Academia serve as network bridges, the Administration Agency's role presents complexities: although it is an essential partner and communication channel, tensions exist between the Administration Agency and AUREMs, stemming from role ambiguity and associated trust deficits. This tension is exemplified by the dispute over whether research permit approval is an administrative task (Administration Agency's view) or a policy-related management aspect (AUREMs' view).

### 5.3. Mismatch 3: Resource constraints and capacity limitations

Resource mismatches severely constrain the capacity of local actors to participate effectively in governance. The effectiveness of co-management is weakened by insufficient investment in capacity and inadequate financial means to support local capacity building. RESEX councils are directly affected by insufficient funding to support members' attendance and engagement.

Furthermore, the influence of the councils was weakened by challenges related to the digitalization deficit, which limited collaboration during periods like the COVID-19 pandemic. These external challenges, including the political-economic crisis, anti-environment governments, and the pandemic, impacted groups' ability to organize and collaborate for integrated management, exacerbating existing governance mismatches.

### 5.4. Synthesis and path forward

Despite the distance from government-related groups, AUREMs have maintained cohesion and initiated integration work via the RESEX Confederation. However, this integration does not effectively include Academia and government-related groups, which need to be mobilized via the councils.

Rejuvenating RESEX councils offers a promising path to addressing these governance mismatches. Strengthened councils could provide platforms to clarify group roles, alleviate co-management tensions, and improve cross-boundary integration. This is crucial because well-defined but adaptable institutional roles are critical for sustainable commons governance, and the councils possess the legal capacity for highly adaptive management, provided they are properly supported and empowered.

This study's methodology—evaluating networks via workshop perceptions combined with interview data and SNA surveys—provides a valuable approach for examining governance networks. By identifying mismatches in RESEX governance, our research contributes to understanding the challenges of implementing co-management arrangements for cross-boundary natural resources.

## Supporting information

**S1 File. Supporting information document.**
(DOCX)

## Acknowledgments

The authors gratefully acknowledge the participants of the interviews and focus groups participants. We also thank the members of LABPEXCA – UFPA for supporting fieldwork activities.

## Author contributions

**Conceptualization:** Rebecca Borges, Theresa Schwenke.

**Data curation:** Rebecca Borges, Theresa Schwenke.

**Formal analysis:** Rebecca Borges, Theresa Schwenke.

**Funding acquisition:** Rebecca Borges.

**Investigation:** Rebecca Borges, Roberta Sá Leitão Barboza.

**Methodology:** Rebecca Borges, Roberta Sá Leitão Barboza, Theresa Schwenke.

**Project administration:** Rebecca Borges.

**Resources:** Rebecca Borges.

**Software:** Rebecca Borges.

**Supervision:** Rebecca Borges, Roberta Sá Leitão Barboza, Theresa Schwenke.

**Visualization:** Theresa Schwenke.

**Writing – original draft:** Rebecca Borges, Roberta Sá Leitão Barboza, Theresa Schwenke.

**Writing – review & editing:** Rebecca Borges, Roberta Sá Leitão Barboza, Theresa Schwenke.

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
