## [Decision Letter · Decision Letter 0]

25 Aug 2025

PONE-D-25-23577Bridges Or Barriers? Cross-boundary communication and Governance Mismatches in Co-Managed Protected AreasPLOS ONE?

Dear Dr. Borges,

Thank you for submitting your manuscript to PLOS ONE. After careful consideration, we feel that it has merit but does not fully meet PLOS ONE’s publication criteria as it currently stands. Therefore, we invite you to submit a revised version of the manuscript that addresses the points raised during the review process.

We look forward to receiving your revised manuscript.

Kind regards,

Umberto Baresi, Ph.D.

Academic Editor

PLOS ONE

Journal Requirements:

[RB, the first author, acknowledges support by the Open Access publication fund of Alfred-Wegener-Institut Helmholtz Zentrum für Polar- und Meeresforschung. The first author also acknowledges the support by the Rufford Foundation.].

[The authors thank the participants of the interviews and focus groups. The first author acknowledges support by the Open Access publication fund of Alfred-Wegener-Institut Helmholtz Zentrum für Polar- und Meeresforschung.]

[RB, the first author, acknowledges support by the Open Access publication fund of Alfred-Wegener-Institut Helmholtz Zentrum für Polar- und Meeresforschung. The first author also acknowledges the support by the Rufford Foundation.]

5. Please amend the manuscript submission data (via Edit Submission) to include author Roberta Barboza.

6. Please amend your authorship list in your manuscript file to include author Roberta Sá Leitao Barboza.

7. We are unable to open your Figure file [Figures.7z]. Please kindly revise as necessary and re-upload.

Additional Editor Comments:

The quality of this paper is being recognized by the reviewers, of which I am extremely pleased.

I encourage the authors to review their manuscript to address the comments provided by the two anonymous reviewers, as I believe that the manuscript would benefit from this.

Reviewers' comments:

Reviewer's Responses to Questions

**Comments to the Author**

1. Is the manuscript technically sound, and do the data support the conclusions?

Reviewer #1: Yes

Reviewer #2: Yes

2. Has the statistical analysis been performed appropriately and rigorously?

Reviewer #1: Yes

Reviewer #2: Yes

3. Have the authors made all data underlying the findings in their manuscript fully available?

Reviewer #1: No

Reviewer #2: Yes

4. Is the manuscript presented in an intelligible fashion and written in standard English?

Reviewer #1: Yes

Reviewer #2: Yes

Reviewer #1: Dear Authors,

Thank you for the opportunity to review your manuscript, which presents an insightful and original investigation into cross-boundary communication and governance mismatches in Brazilian coastal-marine Extractive Reserves (RESEXs). Your use of social network analysis (SNA) in combination with qualitative data offers a rich, multidimensional view of how institutional actors function within and across protected area governance landscapes.

The research is timely, methodologically thoughtful, and conceptually rigorous. It contributes meaningfully to scholarship on co-management, marine spatial planning, and the socio-institutional dimensions of environmental governance—particularly within the under-researched context of community-managed marine protected areas in the Global South.

This is a high-quality manuscript that presents novel, methodologically sound, and theoretically important findings.

To enhance its impact and ensure full compliance with journal requirements, I recommend minor revisions, particularly in relation to:

• Ethical declarations and data availability transparency.

• Methodological detail, particularly regarding interview sampling and SNA metrics.

• Minor enhancements to figures and formatting for accessibility.

I commend your interdisciplinary and participatory approach to understanding marine governance systems, and I am confident that, with these revisions, the manuscript will make a valuable contribution to both academic literature and practical conservation policy.

Thank you again for your important work and for submitting your manuscript to PLOS ONE. I hope these comments are helpful as you prepare your revised submission.

Reviewer #2: Overview:

You provide a thorough case study of select co-managed protected areas in Brazil. Using social network analysis, you reveal a number of challenges with the current co-management governance regime and identify tangible solutions to improve the situation e.g., rebalancing power and increasing funding for participation and influence of local actors. Notably, and contrary to legal intent, your study found that the RESEX councils are not central actors in the networks studies, indicating they are not functioning as intended. You lay out a strong case for your findings through a thorough review of relevant case study material and robust research methods spanning many years. The methods used and the findings articulated are robust and insightful for regional governance improvements and global insights for improving strategies for co-management in protected areas. The main concerns I have are with the readability of the paper overall, including the extensive details provided in the introduction about the case study. See my comments below for suggestions to improve the readability of the manuscript. All told, nice job. A solid applied research effort and many important findings discovered.

Specific Feedback:

35-37. There may be other types of spatial mismatches present in sustainable-use protected areas beyond your example (e.g., park authorities don’t control management of all resource use that affects the park), so I suggest you generalize your statement a bit more to note that you are observing one type. The first second sentence seems to imply that the only type of spatial mismatch possible is the one described. Maybe add “often” or “can” to generalize more. More broadly, you discuss a number of types of mismatches found in your study (1153-1160), but you only highlight spatial in the abstract. I suggest expanding on the types of mismatches found in the abstract. Also consider discussing them in the discussion and not the conclusion. Save the conclusion section for only restating.

82. State the country whose coast you are referring to here. The country of the study has not yet been noted in the introduction.

85. Consider adding more definition and/or examples fr the types of disputes unresolved. Are they “nature resource” only?

88-89. Cite a source here that supports your statement about the intent of RESEXs. I suggest the statute(s) or a reference document from the government. You cite some later but here would be more relevant.

94-95. This sentence is a little confusing. Were these additional rights supported in the past in their legal passage? If so, should it read as “were” supported? If you want to keep it present tense, you may want to modify the sentence at the start. Also, you need a hyphen between “political” and “social.” Alternatively, and more typically, this is written as “socio-political.”

95-98. This is a long sentence with several points. Can you shorten it or make two?

99-101. It would be helpful here to know who those other stakeholders are in general.

104-106. Seems like you are missing a word or two. Before the hyphen in this sentence it does not make complete sense.

78-162. This intro section is good, but I do think it would benefit from a little less detail about your case study (maybe put some of it in the discussion and/or Study Site section?) and add more discussion about what other similar studies have found. I understand yours is the first one to explore connections (bridges) between actors, but I also know there is a lot of literature out there on co-managed protected areas (using network analysis tools?) that would help to better contextualize your research. There must be literature that explores such bridging connections at some level? Overall, the low level of additional context from the literature is a red flag about your due diligence on this topic, so just to overcome that concern alone, I encourage you to provide more literature review. Likewise, you jump into a lot of details about the case here, when this section would benefit from keeping your review of relevant info more general. In the same vein, but at the regional level, you noted in line 256 that a number of studies have been done on the area. Tell us more about them here and in the discussion. What did they find? What are the gaps? How does your study build on this work? Overall, more literature review is needed in the introduction both to contextualize your research topic and research methods, but also relevant regional findings to date. No more than a few more sentences for each would go a long way. Again, also reducing any unnecessary information about the case study to make this part more pithy. It reads as too long currently but also with some serious gaps in reference.

75-76. Should actors be defined here as “who” and not “what”? I suggest: “…Who are the central actors…?”

212-215. Missing a word. The first part of the sentence reads incompletely. If referring to the study (20), typically one would name the first author and et al. For example, “Scullion and others (20), found that…”

219. Define this acronym here, ICMBio. You define it in 249-250, but it should be done here.

226-227. Likewise, define your acronyms.

228-229. Cite your source for this conclusion. Or move it to your discussion as your conclusion.

252-254. Great point to note.

256 & 311. Spell out “SNA” in the titles. No need to abbreviate.

472. I may have missed it but why were regional council representatives not included in the study? Call this out here or above if not already.

521-983. Your results section is long. Is there a way to move some of this material to supporting documentation? Or consolidate it to make it easier to follow? Maybe drop a quote or two? The content is really good, so I am not sure where to cut, but I do think you have too much detail for one paper. Might be worth carving off some of the results for second publication. Overall, this manuscript leans too long in my view. It’s hard to follow with so many elements and details. It would benefit from more structure and fewer details. I can see the value of leaving the details too, so only my opinion here.

983-1143. Great findings. Well described and organized.

1145-1202. Solid conclusions. There are many though and they seem to be in a random order. Can you use sub-titles or some other method to group them in a more synthetic and intuitive order? Again, brings me back to the desire for you to create a more robust structure for organizing your paper throughout. You have great titles and sections, but could they be consolidated a bit more to provide a more consistent and logical flow that unites the details more consistently?

**Do you want your identity to be public for this peer review?** For information about this choice, including consent withdrawal, please see our For information about this choice, including consent withdrawal, please see our Privacy Policy .

Reviewer #1: No

Reviewer #2: No

While revising your submission, please upload your figure files to the Preflight Analysis and Conversion Engine (PACE) digital diagnostic tool, https://pacev2.apexcovantage.com/ . PACE helps ensure that figures meet PLOS requirements. To use PACE, you must first register as a user. Registration is free. Then, login and navigate to the UPLOAD tab, where you will find detailed instructions on how to use the tool. If you encounter any issues or have any questions when using PACE, please email PLOS at . PACE helps ensure that figures meet PLOS requirements. To use PACE, you must first register as a user. Registration is free. Then, login and navigate to the UPLOAD tab, where you will find detailed instructions on how to use the tool. If you encounter any issues or have any questions when using PACE, please email PLOS at figures@plos.org . Please note that Supporting Information files do not need this step.

---

## [Author Response · Author response to Decision Letter 1]

20 Jan 2026

Dear Editor,

Thank you very much for coordinating the publishing process and for your effort in improving our manuscript. We believe your suggestions, as well as the points raised by the reviewers, have contributed to increasing the quality of our paper and therefore its impact.

We hope we have addressed all the points raised in a satisfying manner.

Yours sincerely,

The authors

Specific replies to comments:

and

Reply: We have reviewed the requirements and made the corresponding changes in our manuscript.

[RB, the first author, acknowledges support by the Open Access publication fund of Alfred-Wegener-Institut Helmholtz Zentrum für Polar- und Meeresforschung. The first author also acknowledges the support by the Rufford Foundation.].

Reply: Thank you for stressing the need for completeness of our funding declaration. We adapted it as follows:

"The authors gratefully acknowledge the support by the Open Access publication fund of Alfred-Wegener-Institut Helmholtz Zentrum für Polar- und Meeresforschung (AWI). We extend our sincere thanks to the Helmholtz Institute for Functional Marine Biodiversity (HIFMB) for financially supporting the fieldwork undertaken by the first author. We also express our appreciation to the Rufford Foundation for contributing to the funding part of the fieldwork activities. Authors two and three do not have any additional funding to declare for this research. There was no additional external funding received for this study."

We added it to our cover letter.

[The authors thank the participants of the interviews and focus groups. The first author acknowledges support by the Open Access publication fund of Alfred-Wegener-Institut Helmholtz Zentrum für Polar- und Meeresforschung.]

[RB, the first author, acknowledges support by the Open Access publication fund of Alfred-Wegener-Institut Helmholtz Zentrum für Polar- und Meeresforschung. The first author also acknowledges the support by the Rufford Foundation.]

Reply: Thank you for pointing towards information consistency in our manuscript. We changed our acknowledgement and removed the funding related information. It now reads as follows:

“The authors gratefully acknowledge the participants of the interviews and focus groups participants. We also thank the members of LABPEXCA - UFPA for supporting fieldwork activities.”

We added it to our cover letter.

Reply: We have added the data onto Zenodo (https://zenodo.org/records/17607600).

5. Please amend the manuscript submission data (via Edit Submission) to include author Roberta Barboza.

Reply: This will be done in the resubmission process.

6. Please amend your authorship list in your manuscript file to include author Roberta Sá Leitao Barboza.

Reply: This has been done.

7. We are unable to open your Figure file [Figures.7z]. Please kindly revise as necessary and re-upload.

Reply: Thank you for pointing that out. We zipped the files with a different program and re-uploaded them. If that does not work, we will reach out to the journal for technical support.

Reply: We have revised the citations and references and corrected them throughout the text.

Additional Editor Comments:

The quality of this paper is being recognized by the reviewers, of which I am extremely pleased.

I encourage the authors to review their manuscript to address the comments provided by the two anonymous reviewers, as I believe that the manuscript would benefit from this.

Reply: The authors are grateful for the acknowledgement of their work. We appreciate the effort put into improving our manuscript and making it more impactful for the journal’s audience.

---

## [Editor Report · Decision Letter 1]

22 Jan 2026

Bridges or Barriers? Cross-boundary communication and Governance Mismatches in Co-Managed Protected Areas

PONE-D-25-23577R1

Dear Dr. Borges,

We’re pleased to inform you that your manuscript has been judged scientifically suitable for publication and will be formally accepted for publication once it meets all outstanding technical requirements.

Kind regards,

Umberto Baresi, Ph.D.

Academic Editor

PLOS One
---

## [Editor Report · Acceptance letter]

PONE-D-25-23577R1

PLOS One

Dear Dr. Borges,

I'm pleased to inform you that your manuscript has been deemed suitable for publication in PLOS One. Congratulations! Your manuscript is now being handed over to our production team.

Kind regards,

on behalf of

Dr. Umberto Baresi

Academic Editor

PLOS One